# Comparison of Clinical and Imaginal Features According to the Pathological Grades of Dysplasia in Branch-Duct Intraductal Papillary Mucinous Neoplasm (BD-IPMN) for Personalized Medicine

**DOI:** 10.3390/jpm13010149

**Published:** 2023-01-12

**Authors:** Ji Eun Na, Jae Keun Park, Jong Kyun Lee, Joo Kyung Park, Kwang Hyuck Lee, Kyu Taek Lee

**Affiliations:** 1Department of Medicine, Inje University Haeundae Paik Hospital, Busan 48108, Republic of Korea; 2Department of Internal Medicine, Kangnam Sacred Heart Hospital, Hallym University College of Medicine, Seoul 07441, Republic of Korea; 3Department of Medicine, Samsung Medical Center, Sungkyunkwan University School of Medicine, Seoul 06351, Republic of Korea; 4Department of Clinical Research Design and Evaluation, SAIHST, Sungkyunkwan University, Seoul 16419, Republic of Korea

**Keywords:** optimal pathologic target for surgery, BD-IPMN, low-grade dysplasia, high-grade dysplasia, invasive carcinoma

## Abstract

**Background:** In patients with BD-IPMN, surgical indications have been focused on finding malignant lesions (HGD, high-grade dysplasia/IC, invasive carcinoma). The aim of this study was to compare the preoperative factors that distinguish HGD from LGD (low-grade dysplasia) and HGD from IC to find the optimal pathologic target for surgery according to individuals, considering surgical risks and outcomes. **Methods:** We retrospectively analyzed 232 patients with BD-IPMN diagnosed based on pathology after surgery and preoperative images. The primary outcome was identifying preoperative factors distinguishing HGD from LGD, and HGD from IC. **Results:** In patients with LGD/HGD, a solid component or an enhancing mural nodule ≥ 5 mm (OR = 9.29; 95% CI: 3.3–54.12; *p* < 0.000) and thickened/enhancing cyst walls (OR = 6.95; 95% CI: 1.68–33.13; *p* = 0.008) were associated with HGD. In patients with malignant lesions (HGD/IC), increased serum CA 19-9 (OR = 12.59; 95% CI: 1.81–87.44; *p* = 0.006) was associated with IC. **Conclusions**: The predictive factors for HGD were the presence of a solid component or an enhancing mural nodule ≥ 5 mm and thickened/enhancing cyst walls compared with LGD, and if accompanied by increased CA 19-9, it might be necessary to urgently evaluate the lesion due to the possibility of progression to IC. Based on this finding, we need to find HGD as the optimal pathologic target for surgery to improve survival in low-surgical-risk patients, and IC could be assumed to be the optimal pathologic target for surgery in high-surgical-risk patients because of high morbidity and mortality associated with surgery.

## 1. Introduction

Intraductal papillary mucinous neoplasm (IPMN) is a pancreatic cystic disease with malignant potential and is classified as main-duct (MD)-IPMN, branch-duct (BD)-IPMN, and mixed-type, which shows both characteristics. MD-IPMN is characterized by segmental or diffuse main-duct dilatation, and BD-IPMN is characterized by pancreatic cysts involving the branch duct connecting to the main duct. MD-IPMN and mixed-type IPMN have higher malignant potential compared to BD-IPMN, and surgery is generally recommended at the time of diagnosis. However, BD-IPMN has relatively lower malignant potential, so surveillance is often applied. In BD-IPMN, surgical indications have been determined in consideration of clinical symptoms, imaging findings suggesting malignancy, and changes during surveillance. The surgical indications differ slightly in each guideline (2015 American Gastroenterological Association (AGA) criteria, 2016 Journal of the American Medical Association (JAMA), 2017 Revisions of the International Consensus Fukuoka Guidelines, European Evidence-Based Guidelines on Pancreatic Cystic Neoplasm (EEBGPCN)) [1,2,3,4,5,6]. This is because the proportion of malignant lesions (high-grade dysplasia (HGD) or invasive carcinoma (IC)) has a mean of 31.1% (range, 14.1–47.9%) in resected BD-IPMN patients. Many studies have suggested relevant factors to reduce unnecessary surgery considering the high morbidity of surgery and to improve survival outcomes by performing surgery at the time a malignant lesion is identified [5,7,8,9,10].

Surgery is recommended for malignant lesions with HGD before progression to IC to improve survival outcomes in low-surgical-risk patients because IC has a higher frequency of recurrence [2,10,11,12]. However, because of high comorbidity and mortality following surgery in high-surgical-risk patients, the optimal pathologic target for surgery in high-surgical-risk patients could present itself when IC is suspected, and if HGD is suspected, careful surveillance could be considered. Therefore, it is important to distinguish HGD from low-grade dysplasia (LGD), and HGD from IC, for determining the optimal pathologic target for surgery for each patient. In previous studies, the focus of surgical indications for IPMN was distinguishing between LGD and malignant lesions (HGD/IC) [13]. The objective of our study was to compare the preoperative factors and evaluate whether preoperative factors could distinguish the optimal pathologic target for surgery in patients with BD-IPMN by distinguishing HGD from LGD, and HGD from IC.

## 2. Materials and Methods

### 2.1. Study Design and Participants

This was a retrospective cohort study consisting of patients who received care at Samsung Medical Center, Seoul, Korea. We included surgically resected BD-IPMN patients between January 2000 and May 2019 (n = 241). Of these, to avoid confusing contamination of our study, we excluded nine patients who met any of the following exclusion criteria: (1) preoperative images were mixed-type IPMN (n = 8) and (2) had concomitant ductal adenocarcinoma, not arising from IPMN (n = 1). Finally, 232 patients with surgically resected BD-IPMN were analyzed (Figure 1). The study protocol was reviewed and approved by the Institutional Review Board of Samsung Medical Center (IRB File Number: SMC2020-05-047-001). The requirement for informed consent was waived because only de-identified data routinely collected during hospital visits were used.

### 2.2. Study Variables

BD-IPMN was diagnosed based on pathology after surgery and preoperative images (abdominal computed tomography (CT) and/or magnetic resonance imaging (MRI)).

BD-IPMN was defined pathologically without main-duct involvement. In the preoperative images, cysts > 5 mm in diameter that communicated with the main duct, and without findings of MD-IPMN (segmental or diffuse dilatation of the main pancreatic duct of >5 mm), were defined as BD-IPMN. BD-IPMN was classified into LGD, HGD, and IC based on pathology, and HGD and IC were defined as malignant lesions.

The primary outcome was to find preoperative factors distinguishing HGD from LGD, and HGD from IC. Therefore, the following variables considered in the surgical indication were collected based on hospital records and preoperative radiologic images (CT and/or MRI): age, sex, body mass index (BMI), diabetes mellitus (DM), new-onset DM, weight loss, serum CA 19-9, serum carcinoembryonic antigen (CEA), location, multifocal, size, “high-risk stigmata (HRS)”, “worrisome features (WF)”, endoscopic ultrasonography (EUS) features, and cytology on fine-needle aspiration (FNA). New-onset DM was defined as a diagnosis within two years prior to the date of surgery. “HRS” and “WF” were defined based on 2017 Revisions of the International Consensus Fukuoka Guidelines [5]. Solid components and enhancing mural nodules ≥ 5 mm are known to be difficult to distinguish, so they were set as one variable [14]. EUS features were defined as concerning when there was a definitive mural nodule ≥ 5 mm or main-duct features suspicious of involvement (any one of thickened walls > 2 mm, intraductal mucin or mural nodules). Cytology on FNA was classified by the baseline characteristics low cellularity, negative, atypical cells, and suspicious or positive for malignancy. The baseline characteristic suspicious or positive for malignancy was defined as positive. The secondary outcome was to evaluate whether HGD could be distinguished from LGD, and HGD from IC, by the number of “WF” and/or “HRS”.

### 2.3. Statistical Analysis

Descriptive statistics for the continuous variables and categorical variables are presented as medians (IQR) and frequencies (%), respectively. The baseline characteristics between the three groups were compared using Dunnett’s test and the Bonferroni method, as appropriate. To find preoperative factors that could distinguish HGD from LGD and HGD from IC, the analysis was designed to find factors predicting HGD in patients with LGD/HGD and factors predicting IC in patients with malignant lesions (HGD/IC). For this analysis, univariate and multivariate logistic regression analyses were performed and Bonferroni correction was used to adjust for multiple comparisons. Factors with a *p*-value of <0.05 in the univariate analysis were included in the multivariate analysis. To evaluate whether HGD was predictable by the number of “WF” and/or “HRS” in patients with LGD/HGD, and whether IC was predictable by the number of “WF” and/or “HRS” in patients with malignant lesions, logistic regression analysis was performed and Bonferroni correction was used to adjust for multiple comparisons.

## 3. Results

### 3.1. Comparison of Baseline Characteristics

A total of 232 patients were analyzed, with 181 patients in the LGD group, 24 patients in the HGD group, and 27 patients in the IC group (Figure 1). The baseline characteristics are shown in Table 1. Basic characteristics including age, BMI, DM, new onset of DM, and weight loss were not significantly different among the groups (*p* > 0.05). Serum level of tumor markers (CA 19-9, CEA) did not differ among the three groups (*p* > 0.05). In CT findings, there were also no significant differences in the location, number, and size of the lesions (*p* > 0.05). However, specific variables included in the HRS and WF were significantly different among the study groups. Among the items belonging to HRS, there were significant differences in obstructive jaundice with cystic lesion, and solid component or enhancing mural nodule ≥ 5 mm and ≥ 10 mm among the groups (*p* < 0.05). Additionally, thickened/enhancing cyst walls and increased serum CA 19-9 (≥37 U/mL) showed significant differences (*p* < 0.05). Regarding the features on EUS and cytology on FNA, there were statistically significant differences. Features on EUS included definite mural nodule(s) ≥ 5 mm and suspicious for main-duct involvement, all of which were statistically significant.

### 3.2. Factors That Distinguish HGD from LGD and HGD from IC

The HGD group had a lower proportion of males (*p* < 0.000) and a higher proportion of solid components (*p* < 0.000) or enhancing mural nodules ≥ 5 mm (*p* < 0.000) than LGD. In multivariate analysis in patients with LGD/HGD, female sex (OR = 0.21, 95% CI: 0.07–0.66; *p* = 0.004), solid components or enhancing mural nodules ≥ 5 mm (OR = 9.29, 95% CI: 3.3–54.12; *p* < 0.000), and thickened/enhancing cyst walls (OR = 6.95, 95% CI: 1.68–33.13; *p* = 0.008) were associated with HGD (Table 2).

On the other hand, the IC group had a higher proportion of patients with obstructive jaundice (*p* < 0.014) and increased serum CA 19-9 ((*p* < 0.001) compared with the HGD group. EUS was performed in 93 patients and 45 of those underwent FNA. The HGD group had a higher proportion of patients with positive cytology compared with the LGD group (*p* = 0.013) and the IC group had a higher proportion with concerning features on EUS compared with the HGD group (*p* = 0.015) (Table 1). In multivariate analysis in patients with malignant lesions, increased serum CA 19-9 (OR = 12.59, 95% CI: 1.81–87.44; *p* = 0.006) was associated with IC (Table 3).

### 3.3. Prediction of Malignant Lesion Based on Presence or Absence of “WF” and/or “HRS”

In patients with LGD/HGD, the OR for the risk of HGD tended to increase with increasing numbers of “WF” in patients with no “HRS”; however, it was not statically significant. In patients with “HRS”, the prediction of HGD was not significantly associated with increases in the number of instances of “HRS” (Table 4). In patients with malignant lesions, the prediction of IC was not significantly associated with increases in the number of “WF” or increases in the number of instances of “HRS” (Table 5).

## 4. Discussion

In this study, the HGD group had a higher proportion of solid components or enhancing mural nodules ≥ 5 mm and thickened/enhancing cyst walls compared to the LGD group, and these factors were confirmed as highly relevant in the multivariate analysis. The IC group had a higher proportion of obstructive jaundice and increased CA 19-9 compared to the HGD group, and increased CA 19-9 was a highly relevant factor in the multivariate analysis. This suggests that solid components or enhancing mural nodules ≥ 5 mm and thickened/enhancing of the cyst walls are predictive factors of HGD compared with LGD, and it is likely to progress to IC when accompanied by increased CA19-9. However, an increased CA 19-9 value was found in half of the IC patients, indicating that the CA 19-9 value was less than 37 U/mL in half of the IC patients. Therefore, if a malignant lesion is suspected, accompanied by an increased CA 19-9, it might be necessary to consider surgery for resectable disease even in high-surgical-risk patients. If a malignant lesion is suspected, and the CA 19-9 level is not increased, surgery or careful surveillance might be selected by estimating the surgical risk based on the patient’s age, comorbidities, and surgical procedure.

The proportion of concerning features in EUS was higher in IC compared to HGD, and the proportion of positive cytology was higher in HGD compared to LGD. In case of suspected mural nodules, mural nodules should be defined as hyper-enhancing tissue by using contrast-enhanced EUS (CE-EUS) [15]. Based on these findings, it may be necessary to perform not only CE-EUS but also FNA to find HGD. However, the proportion of concerning features on EUS was 31.3% in HGD and 87.5% in IC, and the proportion of positive cytology was 46.2% in HGD and 60% in IC. Based on these findings, half of the HGD instances may be missed, even after EUS-FNA. Therefore, even if the results of EUS-FNA are not conclusive, if there is a solid component or enhancing mural nodule ≥ 5 mm, thickened/enhancing cyst walls, or increased CA19-9, it might be necessary to consider surgery at the optimal pathologic target for surgery in low-surgical-risk patients. Recently, new EUS tissue sampling methods such as EUS-guided confocal laser endomicroscopy and endoscopic ultrasound-guided through-the-needle biopsy (TTNB) have been introduced and are expected to improve the low diagnostic yield of EUS-FNA [16,17].

In the summary of the various guidelines (Appendix A), MD-IPMN and mixed-type IPMN were recommended for surgery at the time of diagnosis or if there was either jaundice, a mural nodule, or the main pancreatic duct was >10 mm. In contrast, BD-IPMN not only has a heterogeneous surgical indication that includes clinical presentation, image findings suggesting malignancy, and changes during surveillance, but also the indication for EUS-FNA is different, so the unified consensus is unclear.

Previous studies have tried to distinguish malignant lesions (HGD/IC) from LGD as an indication for surgery. However, among malignant lesions, the frequency of recurrence after surgery was as high as 43.3% in IC, compared to 0–17% in HGD [12]. Therefore, considering the outcome after surgery, it is important to distinguish HGD to determine the optimal pathologic target for surgery for low-surgical-risk patients [11,18,19]. Furthermore, because the morbidity of the surgery was as high as 20–25%, while the proportion of malignant lesions after surgery in BD-IPMN patients was low (median, 31.1%; range, 14.4–47.9%) [5,7], the optimal pathologic target for surgery could be when IC is suspected, rather than HGD, in high-surgical-risk patients. Therefore, the significance of the present study is in identifying preoperative factors that distinguish LGD, HGD, and IC for the optimal pathologic target for surgery according to individual patients.

This is the first study to find preoperative factors associated with the optimal pathologic target for surgery by distinguishing the three groups (LGD/HGD/IC). We need to determine the optimal pathologic target for surgery for individuals in consideration of surgical risks and outcomes after surgery. In addition, all relevant variables, including “worrisome features” or “relative indications” as well as “high-risk stigmata” or “absolute indications” suggested in the four guidelines (2015 American Gastroenterological Association (AGA) criteria, 2016 Journal of the American Medical Association (JAMA), 2017 Revisions of International Consensus Fukuoka Guidelines, European Evidence-Based Guidelines on Pancreatic Cystic Neoplasm (EEBGPCN)) were included in the analysis. In the statistical analyses, errors that could occur in multiple comparisons were corrected using the Bonferroni correction.

In previous studies, the major factors predicting malignant lesions in BD-IPMN patients were solid components or mural nodules, main-duct dilatation, cyst size, thickened/enhancing cyst walls, serum CEA, serum CA 19-9, and new-onset DM [13,20]. Based on our study, solid components or mural nodules ≥ 5 mm and thickened/enhancing cyst walls were factors distinguishing HGD from LGD. If increased CA 19-9 is also present, it is important to consider the possibility of progression to IC. Solid components or enhancing mural nodules are commonly known as relevant factors for malignant lesions, and one study reported that 10 mm-sized mural nodules were an indicator for surgery [21]. Thus, we checked whether there are differences in the three groups regarding the presence of solid components or enhancing mural nodules ≥ 10 mm. Compared to 5 mm, the negative predictive value of the 10 mm standard decreased from 86.1% to 84.9%, but the positive predictive value increased from 63.2% to 74.1%. This suggests that it may be more useful in the determination of surgical indications of malignant lesions. Since pancreatic cystic neoplasms have been thought to evolve from LGD to HGD to IC, there was a question about differences in patient age in the three groups. However, there was no difference [22]. It has been reported that increasing numbers of “WF” and/or instances of “HRS” were associated with predicting malignant lesions [23,24]. However, in this study, the prediction of HGD according to the number of “WF” and/or instances of “HRS” in patients in the LGD/HGD group was not significant, and the prediction of IC in patients with malignant lesions was also not significant.

There were some limitations to this study. First, this study was performed retrospectively with the potential for selection bias. Thus, future prospective validation cohort studies are required to confirm our results. Second, we excluded patients with mixed-type BD-IPMN based on the preoperative images in patients with resected BD-IPMN, that is, in patients pathologically diagnosed with BD-IPMN, cases of segmental or diffuse duct dilatation of the main duct due to duct hypertension were excluded based on the preoperative images. Therefore, the importance of main-duct dilatation may have been underestimated in our study. Third, some patients who did not follow the diagnostic process, such as having EUS-FNA, or did not meet surgical indications based on the guidelines were included in the study. Finally, EUS, CE-EUS, and FNA were not performed for all patients included in the study.

Previous studies have focused on finding malignant lesions (HGD/IC) representing surgical indications in patients with BD-IPMN. However, in low-surgical-risk patients, it is important to use the finding of HGD as an indicator of the optimal pathologic target for surgery to improve survival outcomes. Because of the high morbidity and mortality following surgery in high-surgical-risk patients, IC could be used as an indicator of the optimal pathologic target for surgery in those patients. The predictive factors for HGD were solid components or enhancing mural nodules ≥ 5 mm and thickened/enhancing cyst walls, unlike LGD, and if accompanied by increased CA 19-9, it might be necessary to perform an urgent evaluation due to the possibility of progression to IC. Using these preoperative factors, it is possible to predict the optimal pathological target for BD-IPMN surgery, reducing unnecessary surgery, considering the high morbidity of surgery, and improving survival rate. Well-designed randomized prospective studies are needed to support our study results.

## Figures and Tables

**Figure 1 jpm-13-00149-f001:**
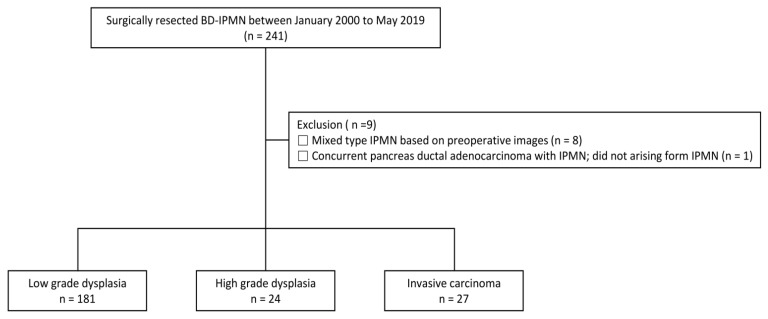
Flow diagram showing the selection for patients on branch-duct intraductal papillary mucinous neoplasm (BD-IPMN).

**Table 1 jpm-13-00149-t001:** Baseline characteristics.

Variables	LGD (n = 181)	HGD (n = 24)	IC (n = 27)	*p*-Value *^,^**
Age, years ^†^	63 (57–69)	67 (55–71)	65 (56–72)	0.768 * 0.948 **
**Sex, male**	126 (69.6)	8 (33.3)	16 (59.3)	**0.000** * 0.064 **
BMI ^†^	24 (22–26)	25 (23–26)	24 (22–25)	0.715 * 0.148 **
DM	36 (19.9)	7 (29.2)	8 (29.6)	0.294 * 0.971 **
New-onset DM	9 (5.0)	1 (4.2)	4 (14.8)	1.000 * 0.354 **
Weight loss	5 (2.8)	0 (0.0)	0 (0.0)	1.000 *
CA 19-9, U/mL ^†^	11 (7–19)	13 (7–21)	39 (6–296)	0.792 * 0.132 **
CEA, ng/mL ^†^	1.6 (1.0–2.5)	1.3 (1.0–1.8)	1.5 (1.1–2.3)	0.304 * 0.389 **
Location, head, neck, and uncinate	112 (61.9)	18 (75)	22 (81.5)	0.210 * 0.574 **
Multifocal	53 (29.3)	8 (33.3)	7 (25.9)	0.683 * 0.562 **
Size based on images, cm ^†^	3.0 (2.3–4.0)	3.5 (3.0–4.5)	3.2 (2.4–4.5)	0.159 * 0.524 **
**High-risk stigmata (HRS)**				
**Obstructive jaundice with cystic lesion**	2 (1.1)	1 (4.2)	10 (37.0)	0.313 * **0.004** **
**Solid component or enhancing mural nodule ≥ 5 mm**	14 (7.7)	9 (37.5)	15 (55.6)	**0.000** * 0.197 **
**Solid component or enhancing mural nodule ≥ 10 mm**	7 (3.9)	8 (33.3)	12 (44.4)	**0.000** * 0.417 **
Main pancreatic duct ≥ 10 mm	0 (0.0)	1 (4.2)	2 (7.4)	0.117 * 1.000 **
**Worrisome features (WF)**				
Pancreatitis	14 (7.7)	2 (8.3)	3 (11.1)	1.000 * 1.000 **
Cyst ≥ 3 cm	108 (59.7)	19 (79.2)	17 (63.0)	0.065 * 0.205 **
Enhancing mural nodule < 5 mm	4 (2.2)	0 (0.0)	0 (0.0)	1.000 *
**Thickened (>2 mm)/enhancing cyst walls**	9 (5.0)	5 (20.8)	6 (22.2)	**0.014** * 0.904 **
Main-duct size 5–9 mm	38 (21.0)	6 (25.0)	4 (14.8)	0.653 * 0.485 **
Abrupt change in duct caliber	0 (0.0)	0 (0.0)	1 (3.7)	1.000 **
Lymphadenopathy	2 (1.1)	0 (0.0)	3 (11.1)	1.000 * 0.238 **
Cystic growth rate ≥ 5 mm/2 years	34 (18.8)	3 (12.5)	3 (11.1)	0.580 * 1.000 **
**Increased serum CA 19-9** **(≥37 U/mL)**	14 (8.0)	2 (8.3)	14 (51.9)	1.000 * **0.001** **
**Features on EUS (n = 93)**	69	16	8	1.000 * **0.015** **
**Definite mural nodule(s) ≥ 5 mm**	21 (30.4)	5 (31.3)	6 (75)	
**Suspicious for main-duct involvement**	4 (5.8)	0 (0.0)	1 (12.5)	
**Cytology on FNA (n = 45)**	27	13	5	**0.013** * 0.268 **
**Low cellularity (inadequate)**	6 (22.2)	2 (15.3)	0 (0.0)	
**Negative**	15 (55.6)	4 (30.8)	0 (0.0)	
**Atypical cell**	5 (18.5)	1 (7.7)	2 (40.0)	
**Suspicious or positive for malignancy**	1 (3.7)	6 (46.2)	3 (60.0)	

HGD, high-grade dysplasia; BMI, body mass index; DM, diabetes mellitus; CA 19-9, carbohydrate antigen 19-9; CEA, carcinoembryonic antigen; EUS, endoscopic ultrasound; FNA, fine-needle aspiration. Values are expressed as n (%) unless otherwise specified. ^†^ Value is median (IQR, interquartile range). * *p*-value was a comparison of Benign and HGD groups and ** *p*-value was a comparison of HGD and invasive carcinoma.

**Table 2 jpm-13-00149-t002:** Factors associated with HGD in patients with LGD/HGD.

Variable	Univariate	Multivariate		
	*p*-Value *	OR	95% CI	*p*-Value *
Sex, male	0.001	0.21	0.07–0.66	0.004
Solid component or enhancing mural nodule ≥ 5 mm	0.001	9.29	3.63–54.12	<0.000
Thickened/enhancing cyst walls	0.029	6.95	1.68–33.13	0.008

OR, odds ratios. * *p*-value was analyzed using univariate and multivariate logistic regression analyses and Bonferroni correction.

**Table 3 jpm-13-00149-t003:** Factors associated with IC in patients with high-risk lesion.

Variable	Univariate	Multivariate		
	*p*-Value *	OR	95% CI	*p*-Value *
Obstructive jaundice with cystic lesion	0.009			
Increased serum CA 19-9 (≥37 U/mL)	0.002	12.59	1.81–87.44	0.006

OR, odds ratios. * *p*-value was analyzed using univariate and multivariate logistic regression analyses and Bonferroni correction.

**Table 4 jpm-13-00149-t004:** Prediction of HGD based on number of “WF” and/or instances of “HRS” in patients with LGD/HGD.

Patients with No “HRS”	LGD/HGD, N	OR	95% CI	*p*-Value
No WF (Ref)	34/0			
1 WF	79/7	6.51	0.23–186.07	0.421
2 WF	40/4	7.67	0.25–236.85	0.366
≥3 WF	12/4	24.85	0.77–806.70	0.077
Patients with “HRS”				
1 HRS only or with one or more WF (Ref)	16/7			
>1 HRS	0/2	11.00	0.14–869.37	0.438

HRS, high-risk stigmata; HGD, high-grade dysplasia; OR, odds ratios; CI, confidence interval; WF, worrisome features; Ref, reference. * *p*-value was analyzed using logistic regression analysis and Bonferroni correction.

**Table 5 jpm-13-00149-t005:** Prediction of IC based on number of presence of “WF” and/or instances of “HRS” in patients with high-risk lesions.

Patients with No “HRS”	HGD/IC, N	OR	95% CI	*p*-Value
No WF	0/0			
1 WF (Ref)	7/4			
2 WF	4/3	1.31	0.14–12.02	1.000
≥3 WF	4/1	0.44	0.03–7.74	1.000
Patients with “HRS”				
1 HRS only or with one or more WF (Ref)	7/12			
>1 HRS	2/7	2.04	0.25–16.50	0.888

HRS, high-risk stigmata; HGD, high-grade dysplasia; IC, invasive carcinoma; OR, odds ratios; CI, confidence interval; WF, worrisome features; Ref, reference. * *p*-value was analyzed using logistic regression analysis and Bonferroni correction.

## Data Availability

The raw data supporting the conclusion of this article will be made available by the authors, without undue reservation.

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
