# Peer review of "Comparison of Clinical and Imaginal Features According to the Pathological Grades of Dysplasia in Branch-Duct Intraductal Papillary Mucinous Neoplasm (BD-IPMN) for Personalized Medicine"

_jpm, 2023, doi:10.3390/jpm13010149_

Round 1

Reviewer 1 Report

This article entitled “Optimal surgery timing of branch-duct intraductal papillary mutinous neoplasm for personalized medicine has been proposed by Na et al and discusses the value of identifying preoperative factors that distinguish HGD from LGD und IC. This study describes an interesting cohort of over 200 patients and highlights the potential use of risk assessment on deciding if a patient should undergo surgery. 

The study is of potential interest, but some aspects should be addressed so it can be considered for publication:

1.      Although this cohort study identified several factors, the analysis does not support the conclusions. There is no data that supports the conclusions about surgery timing: Mortality, morbidity, Survival. Your cohort is interesting on identifying several risk factors but not adequate for defining surgery timing on these patients. I propose a change of the title, objective of the study and conclusions. 

2.      Methods: STROBE Guidelines should be followed and a Checklist provided. Figure 1. : “Patients” not “Articles”.

3. In particular, The discussion should be organized according to these guidelines. Also, Meta-analysis on the topic should be cited. 

4. Please discuss how these findings could influence surveillance on these patients.

Author Response

COVER LETTER

31 Dec 2022

Editor-in-Chief

Journal of Personalized Medicine

Dear Editor-in-Chief

We would like to express our sincere thanks to you and the reviewers for the thorough review of our manuscript (Manuscript ID: jpm-2100594) titled “Optimal surgery timing of branch-duct intraductal papillary mucinous neoplasm (BD-IPMN) for personalized medicine” and for the opportunity to submit a revised and improved version. We believe that by addressing the concerns, we have considerably improved our manuscript. Below this letter, we have provided point-by-point responses to the reviewer’s comment.

We hope that you find the current version of the manuscript suitable for publication in your journal. We will certainly be willing to make additional changes should they be required.

Thank you for your consideration. I look forward to hearing from you.

Sincerely,

Jong Kyun Lee, MD, PhD

Departments of Medicine, Samsung Medical Center, Sungkyunkwan University School

of Medicine, 81 Irwon-ro, Gangnam-gu, Seoul, 06351, South Korea

Email: jongk.lee@samsung.com

+82-2-3410-3409

82-2-3410-6983

Response to Reviewer Comments

Reviewer 1

This article entitled “Optimal surgery timing of branch-duct intraductal papillary mutinous neoplasm for personalized medicine has been proposed by Na et al and discusses the value of identifying preoperative factors that distinguish HGD from LGD und IC. This study describes an interesting cohort of over 200 patients and highlights the potential use of risk assessment on deciding if a patient should undergo surgery. 

The study is of potential interest, but some aspects should be addressed so it can be considered for publication:

Comment 1

Although this cohort study identified several factors, the analysis does not support the conclusions. There is no data that supports the conclusions about surgery timing: Mortality, morbidity, Survival. Your cohort is interesting on identifying several risk factors but not adequate for defining surgery timing on these patients. I propose a change of the title, objective of the study and conclusions. 

Response: Thank you for your feedback. We apologize for the confusion. Our thoughts are in line with you. As we mentioned in the discussion section (PAGE 9) of our study, previous studies have tried to distinguish malignant lesions (HGD/IC) from LGD as an indication for surgery. However, among the malignant lesions, the frequency of recurrence after surgery was as high as 43.3% in IC, compared to 0 – 17% in HGD [12]. Furthermore, because the morbidity of the surgery was as high as 20 –25%, while the pro-portion of malignant lesions after surgery in BD-IPMN patients was low (median, 31.1%; range 14.4 – 47.9%) [5,7]. As shown in Supplementary Fig, therefore, we tried to carefully suggest the timing of surgery according to the risk of surgery for patients in this study. The preoperative factors of our study may help to distinguish HGD from LGD (low grade dysplasia) and HGD from IC to find optimal pathologic target for surgery. We also planned the further large scale multicenter based prospective studies to defining surgery timing on these patients by gathering the data about surgery timing: Mortality, morbidity, Survival.  

Supplementary Fig. Graphical abstract in this study

We revised the title, objective and conclusions of our manuscript as follows;

Revised Text:

Manuscript PAGE 1.

Title section

Comparison of clinical and imaginal features according to the pathological grades of dysplasia in branch-duct intraductal papillary mucinous neoplasm (BD-IPMN) for personalized medicine 

Abstract section 

Background: In patients with BD-IPMN, surgical indications have been focused on finding malignant lesions (HGD, high grade dysplasia/IC, invasive carcinoma). The aim of this study was to compare the preoperative factors that distinguish HGD from LGD (low grade dysplasia) and HGD from IC to find optimal pathologic target for surgery according to individuals, considering surgical risks and outcomes.

Conclusion: The predictive factors for HGD were solid component or enhancing mural nodule ≥ 5 mm and thickened/enhancing cyst walls compared with LGD, and if accompanied by increased CA 19-9, it might be necessary to urgently evaluate the lesion due to the possibility of progression to IC. Based on this finding, we need to find HGD as the optimal pathologic target for surgery to improve survival in the low-surgical-risk patients, and IC could be assumed to the optimal pathologic target for surgery in high-surgical-risk patients because of high morbidity and mortality associated with surgery.

Keywords: Optimal pathologic target for surgery; BD-IPMN; low grade dysplasia; high grade dysplasia; invasive carcinoma

Manuscript PAGE 2.

Introduction section

Line 61-73

Surgery is recommended for malignant lesions with HGD before progression to IC, to improve survival outcomes in low-surgical-risk patients because IC has a higher frequency of recurrence [2,10-12]. However, because of high comorbidity and mortality following surgery in high-surgical-risk patients, the optimal pathologic target for surgery in high-surgical-risk patients could when IC is suspected, and if HGD is suspected, careful surveillance could be considered. Therefore, it is important to distinguish HGD from low-grade dysplasia (LGD), and HGD from IC for determining the optimal pathologic target for surgery for each patient. In previous studies, the focus of surgical indications for IPMN was distinguishing between LGD and malignant lesions (HGD/IC) [13]. The objective of our study was to compare the preoperative factors and evaluate whether preoperative factors could distinguish the optimal pathologic target for surgery in patients with BD-IPMN by distinguishing HGD from LGD, and HGD from IC.

Manuscript PAGE 9.

Discussion section

Previous studies have tried to distinguish malignant lesions (HGD/IC) from LGD as an indication for surgery. However, among the malignant lesions, the frequency of recurrence after surgery was as high as 43.3% in IC, compared to 0 – 17% in HGD [12]. Therefore, considering the outcome after surgery, it is important to distinguish HGD to deter-mine the optimal pathologic target for surgery for low-surgical-risk patients [11,15,16]. Furthermore, because the morbidity of the surgery was as high as 20 –25%, while the proportion of malignant lesions after surgery in BD-IPMN patients was low (median, 31.1%; range 14.4 – 47.9%) [5,7], the optimal pathologic target for surgery could be when IC was suspected, rather than for HGD, in high-surgical-risk patients. Therefore, the significance of the present study is in identifying preoperative factors that distinguish LGD, HGD, and IC for the optimal pathologic target for surgery according to individual patients.

This is the first study to find preoperative factors associated with the optimal pathologic target for surgery by distinguishing the three groups (LGD/HGD/IC). We need to determine the optimal pathologic target for surgery for individuals in consideration of surgical risks and outcomes after surgery.

Manuscript PAGE 10.

Discussion section

Previous studies have focused on finding malignant lesions (HGD/IC) representing surgical indications in patients with BD-IPMN. However, in low-surgical-risk patients, it is important to use the finding of HGD as an indicator of optimal pathologic target for surgery to improve survival outcomes. Because of the high morbidity and mortality following surgery in high-surgical-risk patients, IC could be used as an indicator of the optimal pathologic target for surgery in those patients.

Comment 2

Methods: STROBE Guidelines should be followed, and a Checklist provided.

Figure 1.: “Patients” not “Articles”.

Response: Thank you for your feedback. We apologize for the confusion. Our thoughts are in line with you. First, according to the STROBE Guidelines checklist, we check the Method section of our manuscript as follows 

-  Item 4, 5, 6, 10:

Manuscript PAGE 2.

  1. Materials and Methods

2.1. Study design and participants

This was a retrospective cohort study consisting of patients who received care at Samsung Medical Center, Seoul, Korea. We included surgically resected BD-IPMN patients between January 2000 and May 2019 (n = 241). Of these, to avoid confusing contamination of our study, we excluded nine patients who met any of the following exclusion criteria: (1) preoperative images were mixed-type IPMN (n = 8) and (2) concomitant ductal adenocarcinoma, not arising from IPMN (n = 1). Finally, 232 patients with surgically resected BD-IPMN were analyzed (Fig. 1). The study protocol was reviewed and approved by the Institutional Review Board of Samsung Medical Center (IRB File Number: SMC2020-05-047-001). The requirement for informed consent was waived because only de-identified data routinely collected during hospital visits were used.

- Item 7, 8: We revised the Study variables section of our manuscript as follows to clarify the definition;

Manuscript PAGE 3.

2.2. Study variables

BD-IPMN was diagnosed based on pathology after surgery and preoperative images [abdominal computed tomography (CT) and/or magnetic resonance imaging (MRI)].

BD-IPMN was defined pathologically without main duct involvement. In the pre-operative images, cysts > 5 mm in diameter that communicated with the main duct, and without findings of MD-IPMN (segmental or diffuse dilatation of the main pancreatic duct of > 5 mm) were defined as BD-IPMN. BD-IPMN was classified into LGD, HGD, and IC based on pathology, and HGD and IC were defined as malignant lesions.

The primary outcome was to find preoperative factors distinguishing HGD from LGD, and HGD from IC. Therefore, the following variables considered in the surgical in-dication were collected based on hospital records and preoperative radiologic images (CT and/or MRI): age, sex, body mass index (BMI), diabetes mellitus (DM), new-onset DM, weight loss, serum CA 19-9, serum carcinoembryonic antigen (CEA), location, multifocal, size, “high-risk stigmata (HRS)”, “worrisome features (WF)”, endoscopic ultrasonography(EUS) features, and cytology on fine needle aspiration (FNA). New-onset DM was de-fined as a diagnosis within two years prior to the date of surgery. “HRS” and “WF” were defined based on 2017 Revisions of the International Consensus Fukuoka Guidelines [5]. Solid components and enhancing mural nodules ≥ 5 mm are known to be difficult to dis-tinguish, so they were set as one variable [14]. EUS features were defined as concerning when there was a definitive mural nodule ≥ 5 mm or main duct features suspicious of in-volvement (any one of thickened walls > 2 mm, intraductal mucin or mural nodules). The cytology on FNA was classified as low cellularity, negative, atypical cells, and suspicious or positive for malignancy, and suspicious or positive for malignancy was defined as pos-itive. The secondary outcome was to evaluate whether HGD could be distinguished from LGD, and HGD from IC by the number of “WF” and/or “HRS”.

- Item 9: According to the STROBE Guidelines checklist, the following items have been added to describe any efforts to address potential sources of bias (Item number 9). We revised the Materials and Methods section of our manuscript as follows;

Revised Text:

Manuscript PAGE 2.

Materials and Methods section

This was a retrospective cohort study consisting of patients who received care at Samsung Medical Center, Seoul, Korea. We included surgically resected BD-IPMN patients between January 2000 and May 2019 (n = 241). Of these, to avoid confusing contamination of our study, we excluded nine patients who met any of the following exclu-sion criteria: (1) preoperative images were mixed-type IPMN (n = 8) and (2) concomitant ductal adenocarcinoma, not arising from IPMN (n = 1). Finally, 232 patients with surgically resected BD-IPMN were analyzed (Fig. 1).

- Item 11, 12:

Manuscript PAGE 3.

Materials and Methods section

2.3. Statistical analysis

Descriptive statistics for the continuous variables and categorical variables are pre-sented as median (IQR) and frequency (%), respectively. The baseline characteristics be-tween the three groups were compared using Dunnett’s test and the Bonferroni method, as appropriate. To find preoperative factors that could distinguish HGD from LGD and HGD from IC, the analysis was designed to find factors predicting HGD in patients with LGD/HGD and factors predicting IC in patients with malignant lesions (HGD/IC). For this analysis, univariate and multivariate logistic regression analysis were performed and Bonferroni correction was used to adjust for multiple comparisons. Factors with a p-value of < 0.05 in the univariate analysis were included in the multivariate analysis. To evaluate whether HGD was predictable by the number of “WF” and/or “HRS” in patients with LGD/HGD, and whether IC was predictable by the number of “WF” and/or “HRS” in pa-tients with malignant lesions, logistic regression analysis was performed and Bonferroni correction was used to adjust for multiple comparisons.

Second, we revised the Materials and Methods section (Figure 1.) of our manuscript as follows;

Revised Text:

Manuscript PAGE 3.

Figure 1. Flow diagram showing the selection for patients on branch-duct intraductal papillary mucinous neoplasm (BD-IPMN)

Comment 3

In particular, the discussion should be organized according to these guidelines.

Also, Meta-analysis on the topic should be cited. 

Response: Thank you for raising this point. Our thoughts are in line with you.

First, according to the STROBE Guidelines checklist, we try to organize our manuscript as following order; Key result, Limitation, Interpretation and Generalisability. We revised and added the Discussion section of our manuscript as follows;

Revised Text:

Manuscript PAGE 8.

  1. Discussion

In this study, the HGD group had a higher proportion of solid components or en-hancing mural nodules ≥ 5 mm and thickened/enhancing cyst walls compared to the LGD group, and these were confirmed as highly relevant factors in the multivariate anal-ysis. The IC group had a higher proportion of obstructive jaundice and increased CA 19-9 compared to the HGD group, and increased CA 19-9 was a highly relevant factor in the multivariate analysis. This suggests that solid components or enhancing mural nodules ≥ 5 mm and thickened/enhancing of the cyst walls are predictive factors of HGD compared with LGD, and it is likely to progress to IC when accompanied by increased CA19-9. However, an increased CA 19-9 value was found in half of the IC patients, indicating that the CA 19-9 value was less than 37 U/ml in half of the IC patients. Therefore, if a malignant lesion is suspected, accompanied by an increased CA 19-9, it might be necessary to consider surgery at resectable disease even in high-surgical-risk patients. If a malignant lesion is suspected, while the CA 19-9 level is not increased, surgery or careful surveil-lance might be selected by estimating the surgical risk based on the patient’s age, comorbidities, and surgical procedure.

The proportion of concerning features in EUS was higher in IC compared to HGD, and the proportion of positive cytology was higher in HGD compared to LGD. In case of suspected mural nodule, mural nodule should be defined as hyper enhancing tissue by using contrast-enhanced EUS (CE-EUS) [15]. Based on these findings, it may be necessary to perform not only CE-EUS but also FNA to find HGD. However, the proportion of con-cerning features on EUS was 31.3% in HGD and 87.5% in IC, and the proportion of posi-tive cytology was 46.2% in HGD and 60% in IC. Based on these findings, half of HGD may be missed, even after EUS-FNA. Therefore, even if the results of EUS-FNA are not conclu-sive, if there is a solid component or enhancing mural nodule ≥ 5 mm, thick-ened/enhancing cyst walls, or increased CA19-9, it might be necessary to consider surgery at optimal pathologic target for surgery timing in low-surgical-risk patients. Recently, new EUS tissue sampling methods such as EUS-guided confocal laser endomicroscopy and endoscopic ultrasound-guided through-the-needle biopsy (TTNB) have been introduced and are expected to improve the low diagnostic yield of EUS-FNA [16][17].

In the summary of the various guidelines (Supplementary Table S1), MD-IPMN and mixed type were recommended for surgery at the time of diagnosis or if there was either jaundice, a mural nodule, or the main pancreatic duct was >10 mm. In contrast, BD-IPMN not only has a heterogeneous surgical indication that includes clinical presentation, im-age findings suggesting malignancy, and changes during surveillance, but also the indi-cation for EUS-FNA is different, so the unified consensus is unclear.

Previous studies have tried to distinguish malignant lesions (HGD/IC) from LGD as an indication for surgery. However, among the malignant lesions, the frequency of recur-rence after surgery was as high as 43.3% in IC, compared to 0 – 17% in HGD [12]. There-fore, considering the outcome after surgery, it is important to distinguish HGD to deter-mine the optimal pathologic target for surgery for low-surgical-risk patients [11,15,16]. Furthermore, because the morbidity of the surgery was as high as 20 –25%, while the proportion of malignant lesions after surgery in BD-IPMN patients was low (median, 31.1%; range 14.4 – 47.9%) [5,7], the optimal pathologic target for surgery could be when IC was suspected, rather than for HGD, in high-surgical-risk patients. Therefore, the significance of the present study is in identifying preoperative factors that distinguish LGD, HGD, and IC for the optimal pathologic target for surgery according to individual patients.

In this study, the HGD group had a higher proportion of solid components or en-hancing mural nodules ≥ 5 mm and thickened/enhancing cyst walls compared to the LGD group, and these were confirmed as highly relevant factors in the multivariate anal-ysis. The IC group had a higher proportion of obstructive jaundice and increased CA 19-9 compared to the HGD group, and increased CA 19-9 was a highly relevant factor in the multivariate analysis. This suggests that solid components or enhancing mural nodules ≥ 5 mm and thickened/enhancing of the cyst walls are predictive factors of HGD compared with LGD, and it is likely to progress to IC when accompanied by increased CA19-9. However, an increased CA 19-9 value was found in half of the IC patients, indicating that the CA 19-9 value was less than 37 U/ml in half of the IC patients. Therefore, if a malig-nant lesion is suspected, accompanied by an increased CA 19-9, it might be necessary to consider surgery at resectable disease even in high-surgical-risk patients. If a malignant lesion is suspected, while the CA 19-9 level is not increased, surgery or careful surveil-lance might be selected by estimating the surgical risk based on the patient’s age, comor-bidities, and surgical procedure.

The proportion of concerning features in EUS was higher in IC compared to HGD, and the proportion of positive cytology was higher in HGD compared to LGD. Based on these findings, it may be necessary to perform not only EUS but also FNA to find HGD. However, the proportion of concerning features on EUS was 31.3% in HGD and 87.5% in IC, and the proportion of positive cytology was 46.2% in HGD and 60% in IC. Based on these findings, half of HGD may be missed, even after EUS-FNA. Therefore, even if the re-sults of EUS-FNA are not conclusive, if there is a solid component or enhancing mural nodule ≥ 5 mm, thickened/enhancing cyst walls, or increased CA19-9, it might be neces-sary to consider surgery at optimal surgery timing in low-surgical-risk patients.

Second, we add the Meta-analysis on the topic in the Discussion section of our manuscript. We revised the Materials and Methods section of our manuscript as follows;

Revised Text:

Manuscript PAGE 8.

  1. Discussion

The proportion of concerning features in EUS was higher in IC compared to HGD, and the proportion of positive cytology was higher in HGD compared to LGD. In case of suspected mural nodule, mural nodule should be defined as hyper enhancing tissue by using contrast-enhanced EUS (CE-EUS) [15].

Reference

  1. Lisotti, A.; Napoleon, B.; Facciorusso, A.; Cominardi, A.; Crinò, S.F.; Brighi, N.; Gincul, R.; Kitano, M.; Yamashita, Y.; Marchegiani, G. Contrast-enhanced EUS for the characterization of mural nodules within pancreatic cystic neoplasms: sys-tematic review and meta-analysis. Gastrointestinal Endoscopy 2021, 94, 881-889. e885.

Comment 4

Please discuss how these findings could influence surveillance on these patients.

Response: Thank you for raising this point. Our thoughts are in line with you. As we mentioned in the Discussion section of our study, previous studies have focused on finding malignant lesions (HGD/IC) representing surgical indications in patients with BD-IPMN. If the optimal pathologic target for surgery for BD-IPMN can be predicted using these preoperative factors, we can reduce unnecessary surgery considering the high morbidity of the surgery and improve survival outcomes. We added this point at the end of the Discussion section of our manuscript as follows;

Revised Text:

Manuscript PAGE 10.

Discussion section

Previous studies have focused on finding malignant lesions (HGD/IC) representing surgical indications in patients with BD-IPMN. However, in low-surgical-risk patients, it is important to use the finding of HGD as an indicator of optimal pathologic target for surgery to improve survival outcomes. Because of the high morbidity and mortality following surgery in high-surgical-risk patients, IC could be used as an indicator of the optimal pathologic target for surgery in those patients. Using these preoperative factors, it is possible to predict the optimal pathological target for BD-IPMN surgery, reducing unnecessary surgery considering the high morbidity of surgery and improving survival rate. Well-designed randomized prospective studies are needed to support our study results.

We thank the editor of Journal of Personalized Medicine and the reviewers once again for their constructive feedback. In addition to the revisions provided above, please note that various minor edits were made to correct grammatical errors in our manuscript.

Reviewer 2 Report

This large retrospective study investigate a very important point: the differentiation between low- and high-grade dysplasia in branch-duct IPMN.

1) The main limitation of this study is that it is a surgical series. This is, by definition, a selection bias and the results of your study may not be applicable to the entire population. This point should be added as a limitation.

2) Was EUS, in case of suspected mural nodule, performed with contrast injection? Mural nodule should be defined ad hyper enhancing tissue inside the cyst. Please, discuss this point and add this reference PMID: 34217751

3) What does thickened wall means? How many mm was considered as cut-off to define thickened wall? Was this feature evaluated on cross-sectional imaging or on EUS? Please clarify.

4) The availability of new EUS tissue sampling methods should be mentioned. Doing so cite PMID: 31542380 and PMID: 35451041

Author Response

COVER LETTER

31 Dec 2022

Editor-in-Chief

Journal of Personalized Medicine

Dear Editor-in-Chief

We would like to express our sincere thanks to you and the reviewers for the thorough review of our manuscript (Manuscript ID: jpm-2100594) titled “Optimal surgery timing of branch-duct intraductal papillary mucinous neoplasm (BD-IPMN) for personalized medicine” and for the opportunity to submit a revised and improved version. We believe that by addressing the concerns, we have considerably improved our manuscript. Below this letter, we have provided point-by-point responses to the reviewer’s comment.

We hope that you find the current version of the manuscript suitable for publication in your journal. We will certainly be willing to make additional changes should they be required.

Thank you for your consideration. I look forward to hearing from you.

Sincerely,

Jong Kyun Lee, MD, PhD

Departments of Medicine, Samsung Medical Center, Sungkyunkwan University School

of Medicine, 81 Irwon-ro, Gangnam-gu, Seoul, 06351, South Korea

Email: jongk.lee@samsung.com

+82-2-3410-3409

82-2-3410-6983

Response to Reviewer Comments

Reviewer 2

This large retrospective study investigate a very important point: the differentiation between low- and high-grade dysplasia in branch-duct IPMN.

Comment 1

The main limitation of this study is that it is a surgical series. This is, by definition, a selection bias and the results of your study may not be applicable to the entire population. This point should be added as a limitation.

Response: Thank you for your feedback. Our thoughts are in line with you. We revised the Discussion section of our manuscript as follows; 

Revised Text:

Manuscript PAGE 10.

  1. Discussion section

There were some limitations to this study. First, this study was performed retrospectively with the potential for selection bias. Thus, future prospective validation cohort studies are required to confirm our results. Second, we excluded patients with mixed-type BD-IPMN on the preoperative images in patients with resected BD-IPMN. That is, in patients pathologically diagnosed with BD-IPMN, the cases of segmental or diffuse duct dilatation of the main duct due to duct hypertension were excluded based on the preoperative images. Therefore, the importance of main duct dilatation may have been underestimated in our study. Third, some patients who did not follow the diag-nostic process, such as having EUS-FNA, or did not meet surgical indications based on the guidelines were included in the study. Finally, EUS, CE-EUS and FNA were not performed in all patients included in the study.

Comment 2

Was EUS, in case of suspected mural nodule, performed with contrast injection? Mural nodule should be defined ad hyper enhancing tissue inside the cyst. Please, discuss this point and add this reference PMID: 34217751

Response: Thank you for your feedback. We apologize for the confusion. Our thoughts are in line with you. When a suspected mural nodule or solid component was observed, we performed contrast-enhanced EUS (CE-EUS) with contrast injection. Plus, mural nodule or solid component should be defined as hyper enhancing tissue inside the cyst by using CE-EUS. We revised the Discussion section of our manuscript and added the reference as you mentioned as follows;    

Revised Text:

Manuscript PAGE 8.

  1. Discussion section

The proportion of concerning features in EUS was higher in IC compared to HGD, and the proportion of positive cytology was higher in HGD compared to LGD. In case of suspected mural nodule, mural nodule should be defined as hyper enhancing tissue by using contrast-enhanced EUS (CE-EUS) [15]. Based on these findings, it may be necessary to perform not only CE-EUS but also FNA to find HGD.

Reference

  1. Lisotti, A.; Napoleon, B.; Facciorusso, A.; Cominardi, A.; Crinò, S.F.; Brighi, N.; Gincul, R.; Kitano, M.; Yamashita, Y.; Marchegiani, G. Contrast-enhanced EUS for the characterization of mural nodules within pancreatic cystic neoplasms: sys-tematic review and meta-analysis. Gastrointestinal Endoscopy 2021, 94, 881-889. e885.

Comment 3

What does thickened wall means? How many mm was considered as cut-off to define thickened wall? Was this feature evaluated on cross-sectional imaging or on EUS? Please clarify.

Response: Thank you for raising this point. We apologize for the confusion. Previous studies, the major factors predicting malignant lesions in BD-IPMN patients were solid components or mural nodules, main duct dilatation, cyst size, thickened/enhancing cyst walls, serum CEA, serum CA 19-9, and new-onset DM [13,20]. If it is usually thicker than 2 mm (> 2 mm), it can be defined as a thickened cyst wall. In this study, the study was conducted based on the case of exceeding 2 mm. This definition can be evaluated using EUS as well as radiologic cross-sectional imaging using CT or MRI. As we mentioned in the Materials and Methods section of our study, BD-IPMN was diagnosed based on pathology after surgery and preoperative images [abdominal computed tomography (CT) and/or magnetic resonance imaging (MRI)]. EUS features were defined as concerning when there was a definitive mural nodule ≥ 5 mm or main duct features suspicious of involvement (any one of thickened walls > 2 mm, intraductal mucin or mural nodules). We revised the Materials and Methods section of our manuscript as follows; 

Revised Text:

Manuscript PAGE 3.

  1. Materials and Methods

2.2. Study variables

EUS features were defined as concerning when there was a definitive mural nodule ≥ 5 mm or main duct features suspicious of involvement (any one of thickened walls > 2 mm, intraductal mucin or mural nodules). The cytology on FNA was classified as low cellularity, negative, atypical cells, and suspicious or positive for malignancy, and suspicious or positive for malignancy was defined as positive. The secondary outcome was to evaluate whether HGD could be distinguished from LGD, and HGD from IC by the number of “WF” and/or “HRS”.

Manuscript PAGE 5.

Table 1. Baseline characteristics

Variable:

Thickened (>2 mm)/enhancing cyst walls

Comment 4

The availability of new EUS tissue sampling methods should be mentioned. Doing so cite PMID: 31542380 and PMID: 35451041

Response: Thank you for this comment. Our thoughts are in line with you. We revised the Discussion section of our manuscript as follows; 

Revised Text:

Manuscript PAGE 8.

  1. Discussion section

The proportion of concerning features in EUS was higher in IC compared to HGD, and the proportion of positive cytology was higher in HGD compared to LGD. In case of suspected mural nodule, mural nodule should be defined as hyper enhancing tissue by using contrast-enhanced EUS (CE-EUS) [15]. Based on these findings, it may be necessary to perform not only CE-EUS but also FNA to find HGD. However, the proportion of concerning features on EUS was 31.3% in HGD and 87.5% in IC, and the proportion of positive cytology was 46.2% in HGD and 60% in IC. Based on these findings, half of HGD may be missed, even after EUS-FNA. Therefore, even if the results of EUS-FNA are not conclusive, if there is a solid component or enhancing mural nodule ≥ 5 mm, thickened/enhancing cyst walls, or increased CA19-9, it might be necessary to consider surgery at the optimal pathologic target for surgery in low-surgical-risk patients. Recently, new EUS tissue sampling methods such as EUS-guided confocal laser endomicroscopy and endoscopic ultrasound-guided through-the-needle biopsy (TTNB) have been introduced and are expected to improve the low diagnostic yield of EUS-FNA [16][17].

Reference

  1. Krishna, S.G.; Hart, P.A.; DeWitt, J.M.; DiMaio, C.J.; Kongkam, P.; Napoleon, B.; Othman, M.O.; Tan, D.M.Y.; Strobel, S.G.; Stanich, P.P. EUS-guided confocal laser endomicroscopy: prediction of dysplasia in intraductal papillary mucinous neoplasms (with video). Gastrointestinal Endoscopy 2020, 91, 551-563. e555.
  2. Facciorusso, A.; Kovacevic, B.; Yang, D.; Vilas-Boas, F.; Martínez-Moreno, B.; Stigliano, S.; Rizzatti, G.; Sacco, M.; Arevalo-Mora, M.; Villarreal-Sanchez, L. Predictors of adverse events after endoscopic ultrasound-guided through-the-needle biopsy of pancreatic cysts: A recursive partitioning analysis. Endoscopy 2022, 54, 1158-1168.

We thank the editor of Journal of Personalized Medicine and the reviewers once again for their constructive feedback. In addition to the revisions provided above, please note that various minor edits were made to correct grammatical errors in our manuscript.

Round 2

Reviewer 1 Report

I have no further comments